



# Ecosystem physio-phenology revealed using circular statistics

Daniel E. Pabon-Moreno[1], Talie Musavi[1], Mirco Migliavacca[1], Markus Reichstein[1,2],
Christine Römermann[2,3], and Miguel D. Mahecha[1,2]

[1]Max Planck Institute for Biogeochemistry, Hans–Knoell–Str. 10, 07745 Jena, Germany
[2]German Centre for Integrative Biodiversity Research (iDiv), Deutscher Platz 5e, 04103 Leipzig, Germany
[3]Friedrich Schiller University, Institute of Ecology and Evolution, Philosophenweg 16, D-07743 Jena, Germany

**Correspondence:** Daniel E. Pabon-Moreno (dpabon@bgc-jena.mpg.de)

**Abstract.**

Quantifying responses of vegetation phenology to climate variability is a key prerequisite to predict shifts in how ecosystem dynamics due to climate change. So far, many studies have focused on responses of classical phenological events (e.g. budburst or flowering) to climatic variability for individual species. Comparatively little is known on physio-phenological events such as

the timing of the maximum gross primary production ($DOY_{GPPmax}$). However, understanding this type of physio-phenological phenomena is an essential element in predicting the response of the terrestrial carbon cycle to climate variability. In this study, we aim to understand how $DOY_{GPPmax}$ depends on climate drivers across 52 eddy-covariance (EC) sites in the FLUXNET network for different regions of the world. Most phenological studies rely on linear methods that cannot be generalized across both hemispheres and therefore do not allow for deriving general rules that can be applied for future predictions. Here we

explore a new class of circular-linear (here called circular) regression approach that may show a path ahead. Circular regression allows relating circular variables (in our case phenological events) to linear predictor variables (e.g. climate conditions). As a proof of concept, we compare the performance of linear and circular regression to recover original coefficients of a predefined circular model on artificial and EC data. We then quantify the sensitivity of $DOY_{GPPmax}$ to air temperature, short-wave incoming radiation, precipitation and vapor pressure deficit using circular regressions. Finally, we evaluate the predictive power of the

regression models for different vegetation types. Our results show that the $DOY_{GPPmax}$ of each FLUXNET site has a unique signature of climatic sensitivities. Overall radiation and temperature are the most relevant controlling factors of $DOY_{GPPmax}$ across sites. The circular approach gives us new insights at the site level. In a Mediterranean shrub-land, for instance, we find that the two growing seasons are controlled by different climatic factors. Although the sensitivity of the $DOY_{GPPmax}$ to the climate drivers is very site specific, it is possible to extrapolate the circular regression model across vegetation types. From

a methodological point of view, our results reveal that circular regression is a robust alternative to conventional phenological analytic frameworks. In particular global analyses can benefit, where phase shifts play a role or double peaked growing seasons may occur.



## 1 Introduction

Phenology is the study of the timing of biological events that can be observed either at the organismic level or at the ecosystem
scale (Lieth, 1974). For the latter, phenology is the study of some integral behavior across phenological states of e.g. the inte-
grated canopy reflectance captured by remote sensing (Richardson et al., 2009; Zhang et al., 2003), or ecosystem-atmosphere
CO2-exchange fluxes (Richardson et al., 2010). In the last case we define these processes that integrate plant physiology and
phenology as ecosystem physio-phenology given that related both the uptake of CO2 by photosynthesis and the timing when
plant photosynthesis start (beginning of the growing season), finish (end of the growing season) or reach its maximum potential
(peak of the growing season). At the scale of ecosystems, phenology is influenced by climate conditions but simultaneously
contributes to the regulation of different micro and macro meteorological conditions. Then, phenology influences the temporal
dynamics of land-atmosphere water and energy exchange fluxes. Likewise, the terrestrial carbon cycle is affected by pheno-
logical controls on CO2 uptake and release (Peñuelas et al., 2009).

The eddy covariance technique allows for continuously measuring the exchange of energy and matter between ecosystems
and atmosphere (Aubinet et al., 2012). These measurements are available for several ecosystems around the world through
the FLUXNET network (Baldocchi et al., 2001). The high temporal resolution of most eddy covariance observations (half-
hourly), enables analyzing the seasonality of the exchange of CO2 between ecosystems and the atmosphere in relationship
with meteorological variables (i.e. radiation, temperature, precipitation, as well as with atmospheric humidity) and soil mois-
ture (Migliavacca et al., 2015; Richardson et al., 2010). Specifically, one can monitor the trajectory of gross primary production
(GPP) along the growing season and can derive phenological transition dates such as start and end of the growing season (e.g.
(Luo et al., 2018)), as well as the timing of the maximum gross primary production, hereafter as referred to as $DOY_{GPPmax}$
(Zhou et al., 2016; Peichl et al., 2018; Wang and Wu, 2019).

Understanding how climate variability affects $DOY_{GPPmax}$ is fundamental given that it is the time when plants reach their
maximum potential for CO2 absorption. This optimum state require that several preconditions be achieved during the growing
season and the preceding starvation phase. So far several studies have focused on studying the variability of GPPmax. For
example Huang et al. (2018) reported the increase of GPPmax at global scale during the last decades. The authors found that
the increase is mainly explained by the expansion of croplands, CO2 fertilization and Nitrogen deposition. Zhou et al. (2017)
studied how the variability of annual GPP is influenced by GPPmax and the start and the end of the growing season. They
found that GPPmax better explains the variability of annual GPP than the days of the beginning and end of the growing sea-
son. Bauerle et al. (2012) studied how photoperiod and temperature influence plants photosynthetic capacity. They found that
photoperiod explains better the variability of photosynthetic capacity than temperature. So far, to the best of our knowledge
only one study has focused on understanding the temporal variability of GPPmax. Wang and Wu (2019) used a combination
of satellite, and eddy covariance data to explore how $DOY_{GPPmax}$ is controlled by climatic conditions. The authors reported
that higher temperature advance $DOY_{GPPmax}$, while the influence of precipitation and radiation were biome-dependent. Never-
theless, this study was geographically located in China therefore, a global approach considering several ecosystems across the
whole latitudinal gradient is still missing.





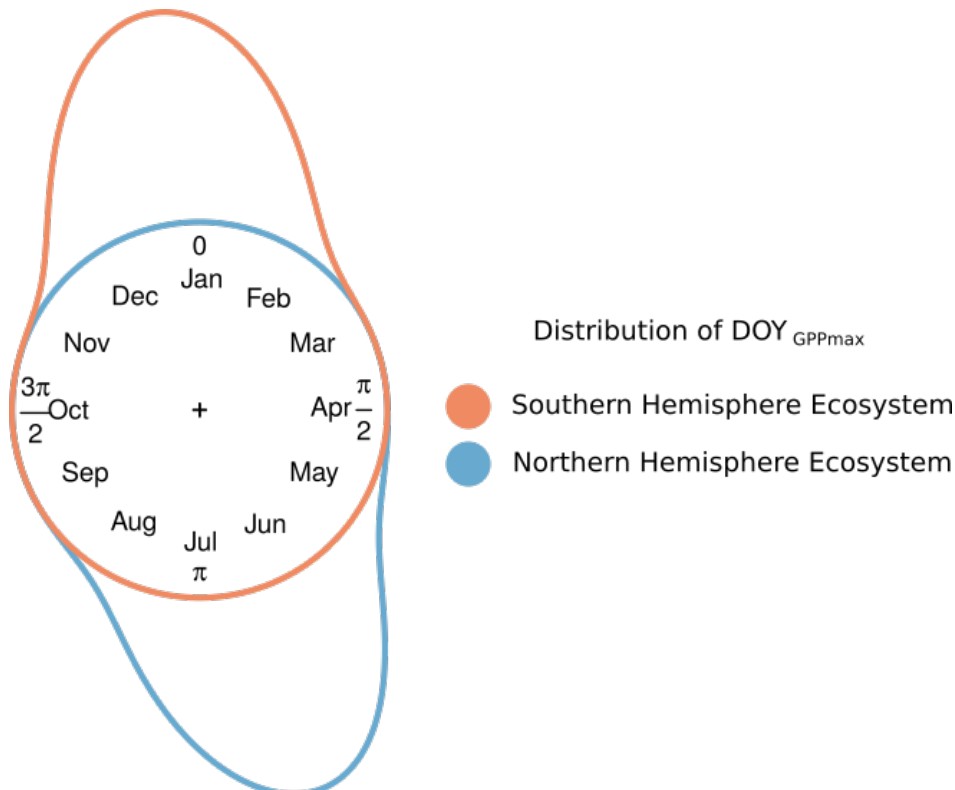

**Figure 1.** Conceptual distribution of GPPmax timing ($DOY_{GPPmax}$) for two hypothetical ecosystems one in the Northern (Blue), and one in the Southern Hemisphere (Red). Each line represents the interannual variability $DOY_{GPPmax}$. DOY= day of the year.

The challenge of understanding phenology is generally to characterize a discrete event recurring with certain periodicity. Classically, phenological analyses have been performed using linear regression models (Morente-López et al., 2018; Rezaei et al., 2018; Zhou et al., 2016). Most of these studies analyze ecosystems with only one growing season (e.g. temperate or boreal

forests), and when the summer is in the middle of the calendar year. However, the existing methods are not sufficiently generic to describe i) ecosystems in the Southern Hemisphere and ii) ecosystems with multiple growing seasons per year as often observed in e.g. semi-arid regions. Figure 1 illustrates the first problem from a conceptual point of view. Assume that some discrete event recurs annually, but the timing varies according to some external drivers. We would then know the interannual variability of phenology which essentially reflects the probability of this recurrent event in the course of the annual cycle.


Figure 1 shows that linear regression models would be inappropriate to predict the day of the year (DOY) of some phenological event in the Southern Hemisphere, as the actual target values to predict may flip between $\gtrsim \frac{3\pi}{2}$ and $\lesssim \frac{\pi}{2}$. In recent years, circular statistics have gained some attention as they offer a solution to problems of this kind (Morellato et al., 2010; Beyene et al., 2018). Unlike classical statistics, the predicted variables are expressed in terms of angular directions (degrees or





radians) across a circumference (Fisher, 1995) allowing to perform statistical analysis where the data space is not Euclidean. In this framework, point events can be described as a von-Mises distribution (Von Mises, 1918) (the equivalent to the normal distribution in circular statistics, as shown in 1) with two parameters: The mean angular direction ($\mu$) and the concentration parameter ($\kappa$). Circular-linear (here called circular) regression technique allows to predict such circular responses (e.g. the timing of phenological events) from other linear variables (Morellato et al., 2010). Given that any phenological event can be

interpreted as an angular direction and modeled alike, we assume that these circular regressions are well suited in this context. Despite this evident suitability, circular statistics have not yet been extensively applied in the study of phenology and will therefore be presented here as an alternative to conventional linear techniques.

In this paper, we aim to identify the factors controlling the phenology of the maximal seasonal GPP (GPPmax). Specifically, we want to understand what are the climate controls of the timing of GPPmax ($DOY_{GPPmax}$) and provide a predictive framework

using circular statistics. We explore this physio-phenological characteristic across different ecosystems around the globe using the FLUXNET 2015 dataset (Pastorello et al., 2017). The questions that we want to answer are: can circular statistics describe and predict $DOY_{GPPmax}$ per vegetation types? Can $DOY_{GPPmax}$ be explained using the climate conditions as cumulative factors? How is $DOY_{GPPmax}$ affected by the climatic conditions during the growing season? Based on these findings we discuss the potential of circular regressions beyond this specific application case in related phenological problems.

## 2 Methods

### 2.1 Data

We use 52 FLUXNET sites (with at least seven years of data) located through the latitudinal gradient of the globe (i.e. Northern, Southern hemisphere and tropical region) from the FLUXNET-2015 database (Table A1, http://fluxnet.fluxdata.org/ (Pastorello et al., 2017)). Each FLUXNET site is identified with an abbreviation of the country and the name of the place e.g. AU-

How means tower in Howard Springs, Australia. From the dataset we use the GPP data that was derived using the nighttime partitioning method and considering the variable u*-t threshold to discriminate values of insufficient turbulence (Reichstein et al., 2005). In order to identify maximum daily GPP, we compute the quantile 0.9 for each day based on the half-hourly flux observations. As potential explanatory variables for $DOY_{GPPmax}$ we use on air temperature (Tair), shortwave incoming radiation (SWin), precipitation (Precip), and vapor pressure deficit (VPD).

Given that the past climate conditions affect the $CO_2$ exchange between the atmosphere (ecological memory, (Liu et al., 2019; Ryan et al., 2015)), we need to understand whether an aggregated form of these climatic variables would better predict the phenological responses. For this we aggregate the original times-series of the Tair, SWin, Precip, and VPD using a half-life decay function (equation 1).

$$mean(x, N) = \frac{\sum_{i,t=1}^{365} x_i N_t}{\sum_{i,t=1}^{365} N_t} \tag{1}$$





where: $N(t) = N_0 e^{\frac{ln(2)}{t_{1/2}}}$


We can then vary the half-time parameter ($t_{1/2}$) from 2 to 365 days. We make these variables comparable via centering standardization to unit variance and identify the optimal $t_{1/2}$ (S1.1).

## 2.2 Circular statistics

A basic circular regression model was proposed by (Fisher and Lee, 1992) as follows:

$\quad y = \mu + 2 * \mathrm{atan}(\beta_i x_i)$ \hfill (2)

where $y$ is the target variable (i.e. $\mathrm{DOY_{GPPmax}}$), $\mu$ is the mean angular direction of the target variable, $x_i$ are the predictor variables, and $\beta_i$ the regression coefficients. The parameters $\mu$ and $\beta_i$ are fitted via the maximum likelihood method using reweighted least squares algorithm as proposed by (Green, 1984).

Circular regression models allow to interpret 1) the sign of the coefficient, 2) the statistical significance of the coefficient, and
3) the accuracy of the prediction. Regarding the first point: Consider a negative sign of the coefficient, this would mean that an increasing value of the predictor would lead to an earlier $\mathrm{DOY_{GPPmax}}$ compared to the mean angular direction. Obviously the inverse would happen when the coefficient is positive. Figure 2 conceptually illustrates how the coefficients affect the predictions. Regarding the second aspect we can state that, if a coefficient is not significant, then its contribution would not be relevant to explain the phenological observation. In our case we define that the coefficient is significant if the median of the
distribution of p-values is less than 0.05. Finally, we can estimate the accuracy of the prediction using the Jammalamadaka-Sarma (JS) correlation coefficient (Jammalamadaka and Sarma, 1988) implemented in the R package "circular" (Agostinelli and Lund, 2017). As in any other regression framework, this approach helps us to quantify the effect of each climate variable on the inter-annual variability of $\mathrm{DOY_{GPPmax}}$.

To estimate the relative sensitivity of $\mathrm{DOY_{GPPmax}}$ to Tair, SWin, Precip, and VPD we use the implementation of equation
2 in the "circular" R package (Agostinelli and Lund, 2017). To increase the robustness of the method we implement a block bootstrapping per growing season generating a model parameter average based on 1000 iterations. In each analysis, we estimate the accuracy of the model using the JS correlation coefficient.

## 2.3 Circular vs Linear Regression

We use equation 2, where we predefined two coefficient regressions ($\beta_1 = 0.3$, $\beta_2 = 0.1$). We generate two scenarios: 1) when
the target timing occur at the beginning of the year ($\mu = 0$) and 2) when the target timing occur at the middle of the year ($\mu = \pi$). We simulate the variables $x_1$ and $x_2$ as normal distributions with a mean of 0, and 4 respectively, and a standard deviation of 1. For each scenario the amount of data is given by the equation 3 where $N$ (rounded) is the amount of data for $x_1$ and $x_2$ and $x$ take arbitrary values from 5 to 1000.

$N = e^{\log(x)}$ \hfill (3)




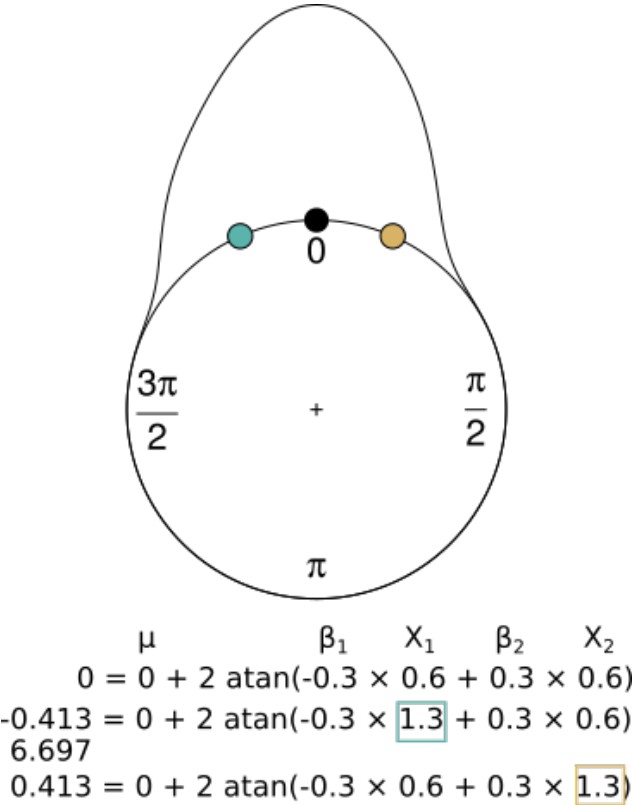

**Figure 2.** Interpretation of the coefficients in the circular regression. Consider a reference point (Black) generated with a circular-linear model with mean angular direction ($\mu = 0$), two coefficients ($\beta_1$,$\beta_2$) and two variables (X1, X2), where one of the coefficients is negative ($\beta_1$) and the other one is positive ($\beta_2$). When the coefficient is negative and the value of the parameter increases (blue) the result is an earlier observation compared with the reference point (The equivalent of the negative radian is shown below the equation). On the other hand, when the coefficient is positive and the variable increase (yellow) the observation is later.

We use the simulated data from equation 1, and the original values of $x_1$ and $x_2$ to recover the original values of the regression coefficients $\beta_1$ and $\beta_2$ using the circular and linear regression. To increase the robustness of the analysis we simulate $x_1$ and $x_2$ 1000 times for each $n$. We estimate the difference between the recovered and the original coefficient as the efficiency of the model (i.e. lower values mean higher efficiency).

### 2.4   Analysis setup

The target variable DOY$_\text{GPPmax}$ is the day of the year when GPP reaches its maximum during the growing season. Given that different ecosystems present more than one growing season per year (e.g. semi-arid ecosystems) it is necessary to identify the number of growing seasons per year. To identifying the number of growing seasons we apply a Fast Fourier Transformation (FFT) (Cooley and Tukey, 1965) to the mean seasonal cycle of the GPP time series. The number of growing seasons is





equal to the maximum absolute value of the first four FFT coefficients (excluding the first one). For each FLUXNET site,
we reconstructed the GPP time series taking the real numbers of the inverse FFT. We used these reconstructed time series to
calculate the expected mean timing of $DOY_{GPPmax}$ and use this value as a template). To recover the real $DOY_{GPPmax}$ from the
original time series we define a window around the template of length inversely proportional to the number of cycles (180 days
/ Number of growing seasons). Finally, to increase the robustness of the analysis we identify the days with the 10 greatest GPP
values. Finally, given that most of the sites are located in the northern hemisphere we expect that in most cases $DOY_{GPPmax}$
will be reached at the middle of the year.

To understand possible similarities in the regression coefficients across sites, and if these are related to the vegetation types
or climate classes, we visualize the coefficients in a reduced dimensional space. For this dimensionality reduction we use
t-Distributed Stochastic Neighbor Embedding (t-SNE) analysis (Maaten and Hinton, 2008) using the "dimRed" R package
(Kraemer et al., 2018). To quantify the contribution of each climate variable, we count the number of sites per vegetation type
where the regression coefficient is statistically significant. We perform a one-leaf-out cross validation per vegetation type to
evaluate the predictive power of the circular regression using climate conditions. We only consider vegetation types with more
than five sites. In this case the standardization of the climate variables is not applied. Finally, we use the mean of the optimum
half-time parameter per vegetation type to weigh the climate conditions.

## 3    Results

Here, we first report results from simulated data to describe the performance of the circular regression approach compared to
a linear model. Second, we compare the performance of circular and linear regression using empirical data. Third, we analyze
the sensitivity of $DOY_{GPPmax}$ across vegetation types and climate classes. Finally, we show the results of the predictive power
of circular regression per vegetation type.

### 3.1    Circular vs Linear Regression

Figure 3 shows that for $\mu = 0$ ($DOY_{GPPmax}$ at the beginning of the year) and $\mu = \pi$ ($DOY_{GPPmax}$ at the middle of the year)
the circular regression method is generally more efficient as it has a lower distance in case of $\beta_1$. For $\beta_2$ linear regression
performs better than circular regression when the number of data is higher than 100. Nevertheless, the differences between
both regressions for $\beta_2$ are of the order of 0.01 while the differences for $\beta_1$ are of the order of 0.1.

To illustrate the method in practice, we compare the circular and linear models using data from two sites: US-Ha1 (Northern
Hemisphere deciduous Broadleaf forest), and AU-How (Southern Hemisphere woody savanna). We relate the climate variables
with $DOY_{GPPmax}$ (See methods) and reconstructed the $DOY_{GPPmax}$ using the linear and circular regression models. We compare
observed and predicted $DOY_{GPPmax}$ using JS correlation for circular model and Pearson-Product Moment for linear model.
For US-Ha1 both methods shows similar performance predicting $DOY_{GPPmax}$ (Figure 4), while for AU-How circular model
recover better the original data than the linear model. In the case when $DOY_{GPPmax}$ is reached at the beginning of the year,
linear methods produce a strong bias predicting the timing across all year (Figure 4,b).





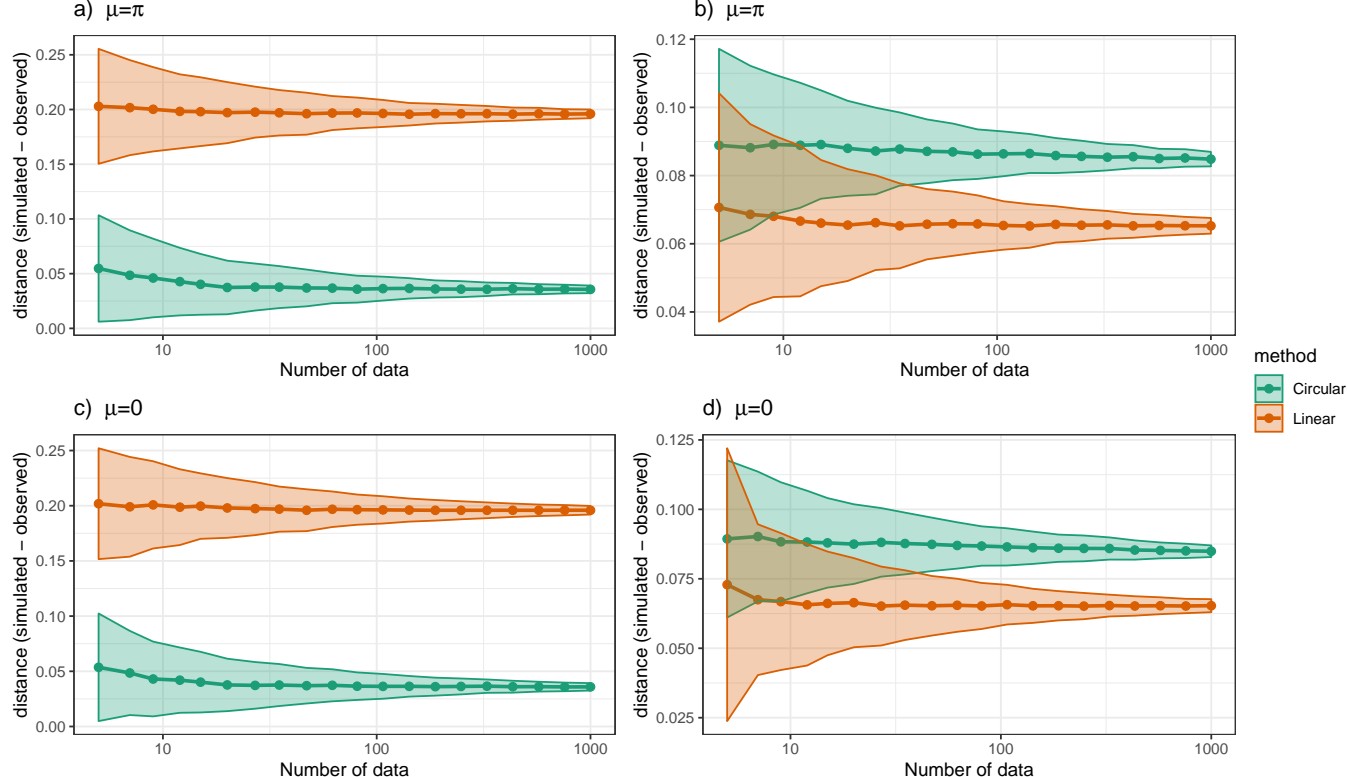

**Figure 3.** Efficiency of linear and circular regression models recovering the original coefficients of a circular regression to the number of data (lower values mean higher efficiency). Upper side: $\mu = \pi$ (Maximum at the middle of the year). Bottom side $\mu = 0$ (Maximum at the beginning of the year). The effect is analyzed for each regression coefficient individually. a. and c. correspond to the regression coefficient $\beta_1$ and b. and d. correspond to the regression coefficient $\beta_2$.

## 3.2 Sensitivity of $DOY_{GPPmax}$ to climate variables

From 52 sites analyzed in this study, only one site (ES-LJu) shows a bimodal growing seasons (see S1.2). As expected in most cases $DOY_{GPPmax}$ occurs at the middle of the calendar year (Figure S6), reflecting the uneven site distribution in FLUXNET (Schimel et al., 2015). However some ecosystems in the Northern Hemisphere do reach $DOY_{GPPmax}$ at the beginning of the

year, these are Mediterranean sites such as, US-Var and ES-LJu. In general terms, most of the sites have a standard deviation between 10 [days] and 40 [days]. The maximal std is 46.9 [days] for AU-Tum site. A detailed table with the mean angular direction and standard deviation of $DOY_{GPPmax}$ of each site is presented in Supplement 1.2.

For most of the sites, the JS correlation coefficients are between 0.98 and 0.85 (Figure S5) showing that the interannual variability of $DOY_{GPPmax}$ is mainly explained by the cumulative effect of the climate variables. Only five sites have a JS

coefficient less than 0.8: US-Ton, IT-MBo, IT-Ro2, US-Wkg, and BR-Sa1. For ES-LJu the JS coefficient for the first growing season is 0.94 and 0.93 for the second one (Table S2).





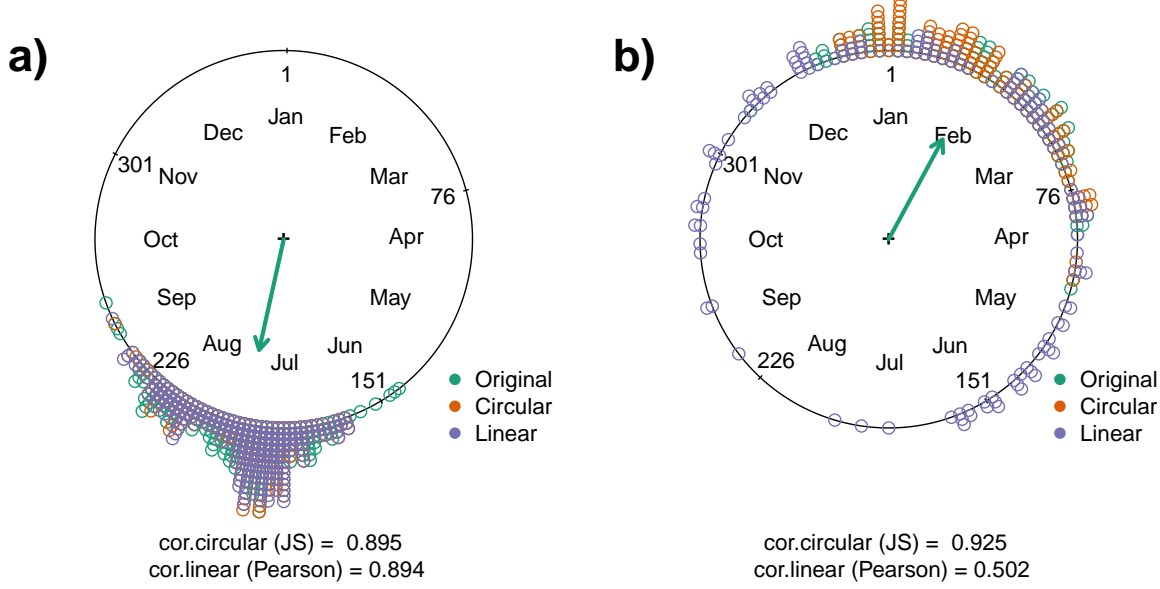

**Figure 4.** Correlation coefficient between the observed and predicted DOY$_{GPPmax}$ using climatic variables. Two sites are presented: a. US-Ha1, and b. AU-How. Observed DOY$_{GPPmax}$ (Green) is compared with the data recovered using Circular (Orange) and Linear (Purple) regressions. Two correlation coefficients are used: Jammalamadaka-Sarna (JS) and Pearson product-moment (Pearson). In the circular plot the months and the day of the year (DOY) are also represented every 75 days. The green arrow indicate the mean angular direction of the distribution.

Across all sites we find that shortwave incoming radiation appears as the dominant driver worldwide in 34 sites (66 %). Air temperature is the main driver at another 14 sites (27 %), while precipitation is the main driver for US-Wkg and VPD for AU-How. For one site (IT-So1) any climatic variable is significant. In terms of the sign of the coefficients, shortwave incoming
radiation and precipitation are predominantly negative, while for VPD is predominantly positive (Table 1). This means that higher integrated values of radiation and precipitation lead to an earlier DOY$_{GPPmax}$, while and increase of VPD will lead to a later DOY$_{GPPmax}$. For air temperature we find no clear tendency, as its signs are almost equally distributed between positive and negative (Table 1). Individual sensitivities per site are shown in Supplement 2.

Each site shows a unique DOY$_{GPPmax}$ sensitivity to the different climate variables which leads to a range of unique patterns
(Figure S7). In fact, these patterns of regression coefficients do not show any systematic relationship with vegetation type or climate class where the ecosystem is located (Figure S7). Considering the frequency per vegetation type, shortwave incoming





**Table 1.** Number of FLUXNET sites where each regression coefficient is statistically significant to explain the phenology of GPPmax ($DOY_{GPPmax}$), and if the coefficient is positive or negative. We each category we present the number of sites. SWin = Shortwave incoming radiation, Tair = Air temperature, Precip = Precipitation, VPD = Vapor pressure deficit.

|  | Climatic variable | | | |
| --- | --- | --- | --- | --- |
| Sign | SWin | Tair | Precip | VPD |
| (+) | 1 | 21 | 2 | 17 |
| (-) | 48 | 17 | 21 | 5 |

radiation has the highest frequency in Evergreen Needleleaf Forest, Deciduous Broadleaf Forest, Grassland, Mixed Forest (MF), and Evergreen Broadleaf Forest, (Figure 5). VPD is not significant for Permanent Wetlands (WET) and Open Shrublands (OSH). While for Closed Shrublands (CSH), and Savannas (SAV) all the climate variables have the same frequency.

A special case to understand the sensitivity of $DOY_{GPPmax}$ to climate variables is the site: "Llano de los Juanes", Spain (ES-LJu). It is the only clearly bimodal ecosystem in our study (Figure 6. In this case nighter SWin nor Precip are statistically significant. While Tair and VPD are significant for both seasons. Furthermore, in the first growing season air temperature has a positive coefficient, while in the second growing season air temperature has a negative sign. On the other hand, VPD has a negative sign (the inverse of temperature) during the first growing season and positive during the second one.

The leave-one-site-out cross-validation for several vegetation types shows that the power of the prediction of the model for GRA is zero. For DBF and EBF is 0.49 and 0.19, respectively, while for MF and ENF the power prediction of the model is 0.68 and 0.7, respectively (Figure 7).

## 4    Discussion

### 4.1    Circular vs Linear regression

We show that circular regression is a suitable tool to analyze phenological events. Our results suggest that circular regressions can recover the values of the predefined coefficients in the simulations with higher accuracy than linear regression (in the order of 0.1 to 0.01), presenting an advantage when analyzing the effect of climatic variables on phenological events. In addition, circular regression is able to analyze the phenology of ecosystems regardless of the day of the year when the event occurs, allowing to analysing phenological studies at global scale regardless of geographic location or the distribution of the

observations during the year.

Richardson et al. (2013) concluded that phenology models need to be improved as a prerequisite to extending the prediction capacity of global-scale models. As we demonstrate here, circular statistics open new opportunities to do so. Besides, the results on phenological sensitivity of $DOY_{GPPmax}$ in this study indicate the complexity of ecosystem responses to climate variability. This should be considered a first step to implement more complex statistical techniques like decision trees, Gaussian process

or artificial neural networks.




**Figure 5.** Contribution of each climate variable to explain the interannual variation of $DOY_{GPPmax}$ per vegetation type. CSH: Closed Shrublands (n = 1), DBF: Deciduous Broadleaf Forest (n = 10), EBF: Evergreen Broadleaf Forest (n = 5), ENF: Evergreen Needleleaf Forest (n = 15), GRA: Grassland (n = 8), MF: Mixed Forest (n = 5), OSH: Open Shrublands (n = 1), SAV: Savannas (n = 1), WET: Permanent wetlands (n = 2), WSA: Woody Savannas (n = 3). Each bar shows the cumulative number of sites where each climate variables are statistically significant.


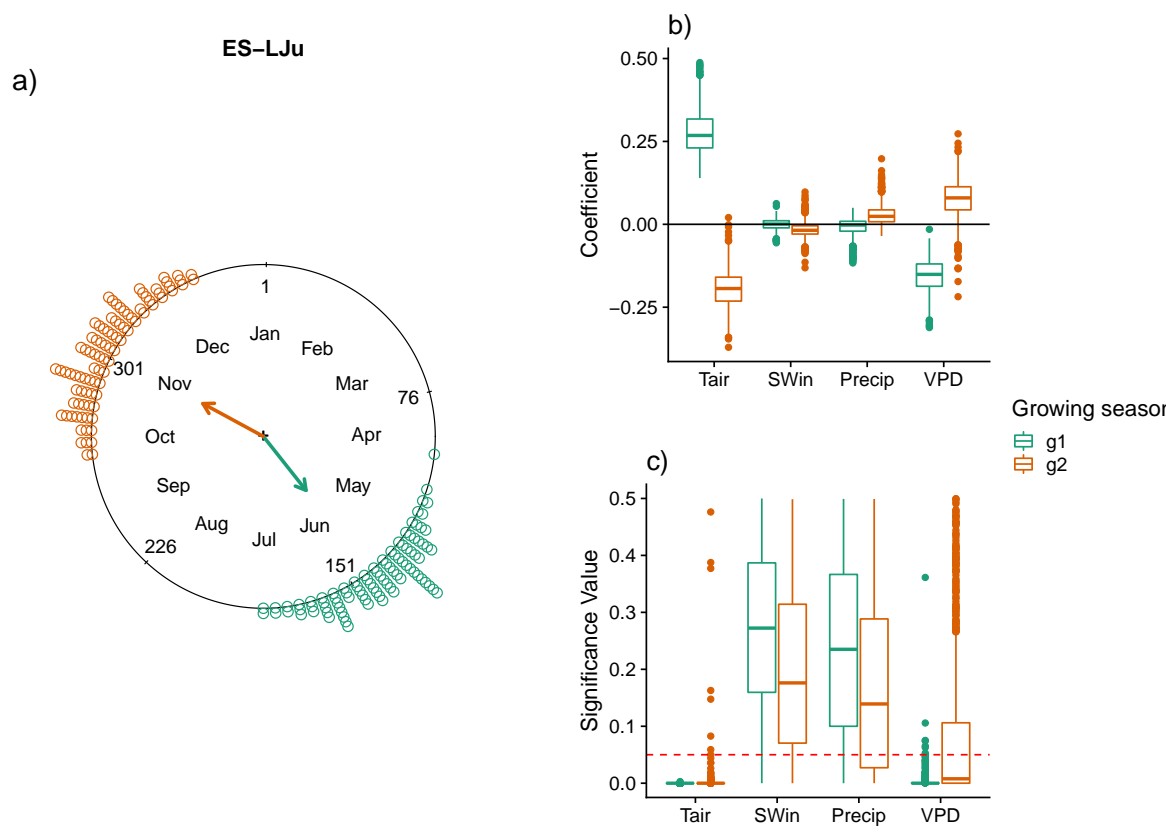

**Figure 6.** DOY$_{GPPmax}$ sensitivity to different climate drivers in a Mediterranean ecosystem: "Llano de los Juanes", Spain (ES-Lju) with two growing seasons (green and orange). a) DOY$_{GPPmax}$ distribution across the year. The arrows indicate the mean angular direction of the growing season. b) regression coefficients for each growing season and c) the significance values for each variable. The red line in c) represents a p-value of 0.05.



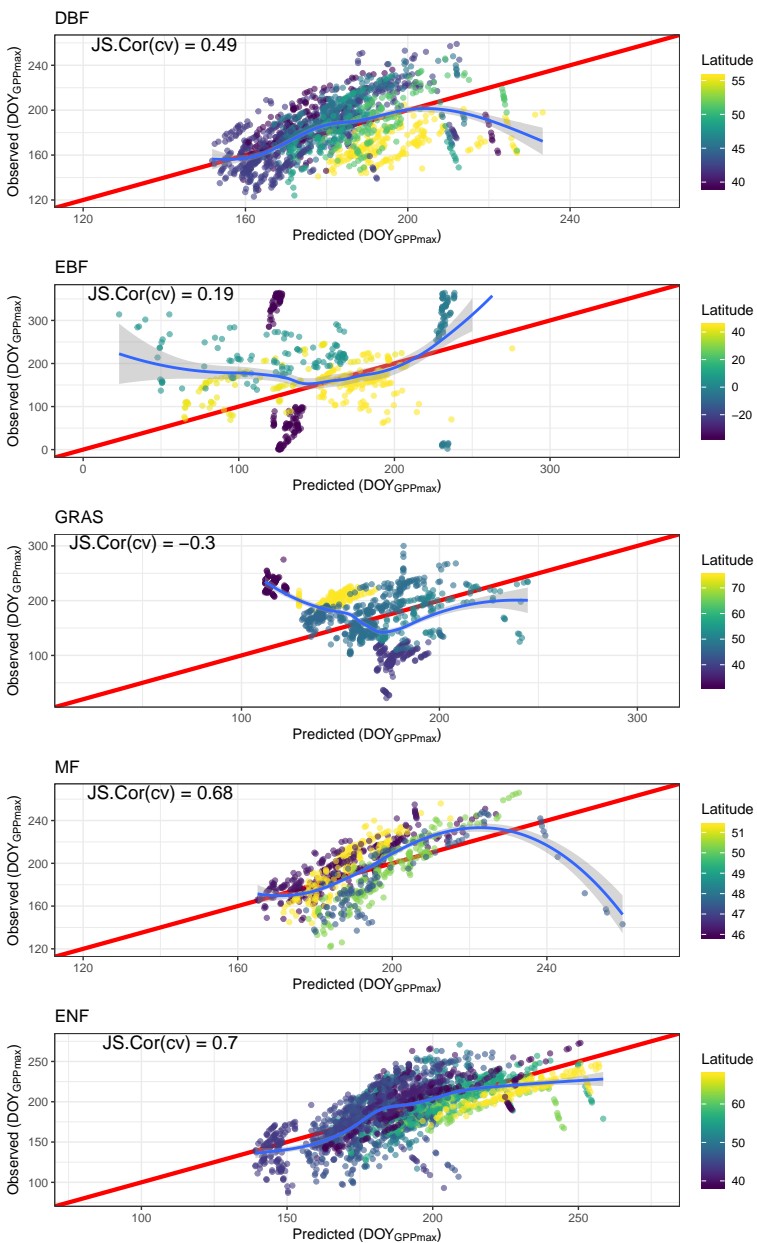

**Figure 7.** Cross validation of the circular regression model to predict DOY$_{GPPmax}$ for different vegetation types using air temperature, Short-wave incoming radiation, precipitation and Vapor pressure deficit (see methods). Deciduous Broadleaf Forest (DBF). Evergreen Broadleaf Forest (EBF). Grassland (GRA). Mixed Forest (MF), and Evergreen Needleleaf Forest (ENF). For each site the Jammalamadaka-Sarna (JS) correlation coefficient is shown. The red line represents the perfect fit. The blue line shows the tendency of the data.





## 4.2 Sensitivity of $DOY_{GPPmax}$ to climate variables

The geographical location of the FLUXNET 2015 sites represent an advantage to capture the $DOY_{GPPmax}$ variability at global scale (Figure S6). Most of the analyzed sites (47) are located in the Northern Hemisphere. Two sites (GF-Guy and BR-Sa1) are located in the tropical region and, 3 sites (ZA-Kru, AU-How, AU-Tum) in the Southern Hemisphere. However, because

of the low number of sites reported in the tropical and southern region with more than seven years of data, our understanding about the $DOY_{GPPmax}$ variability in these regions is still limited. For that, increasing the data available for tropical and southern regions should be a fundamental task during the next decade to complement our knowledge about the physio-phenological ecosystem state.

The high values of the JS correlation coefficient for most of the sites demonstrate that the interannual variability of $DOY_{GPPmax}$

can be explained as the cumulative effect of the climate variables during the growing season. Sites where it was not possible to explain the variations of $DOY_{GPPmax}$ with enough confidence level (JS correlation < 0.8) might need an incorporation of biotic variables (e.g. species composition (Peichl et al., 2018)) that can improve the power prediction of the model.

Our results suggest that there is no pattern between the $DOY_{GPPmax}$ sensitivity across vegetation type or climate classes. In other words, the $DOY_{GPPmax}$ sensitivity is site-specific, probably produced by the unique combination of biotic (e.g. species

composition, species dominance, species phenology, species interaction, and phenotypic plasticity) factors that are not evaluated in our study. Several studies that focussed on ecosystem phenology suggest that species composition play a fundamental role in ecosystem phenology of the CO2 uptake (Gonsamo et al., 2017; Peichl et al., 2018). Nevertheless, our results show that the interannual variability of $DOY_{GPPmax}$ is still climatically driven.

While there is no clear relationship between the $DOY_{GPPmax}$ sensitivity and the vegetation type, we find a predominant role of Shortwave incoming radiation (SWin) at the global scale on the $DOY_{GPPmax}$ interannual variability, where for most of the sites SWin has a negative regression coefficient. This means, that if the SWin increases during the growing season the $DOY_{GPPmax}$ will be reached earlier. This SWin effect can be a consequence of $DOY_{GPPmax}$ being reached at the same time as SWin is maximum.

The second predominant factor at global scale is air temperature (Tair). However, there is not a clear pattern in the sign of the regression coefficient (positive or negative) at global scale. Our hypothesis is that the sign of Tair is reflecting the speed consumption of the water available in the soil (water budget). In this way when the regression coefficient is positive and Tair increases during the growing season the $DOY_{GPPmax}$ will be reached later reflecting a decrease in the speed of water consumption, and increasing the length of the growing season (Figure 8). Several studies demonstrated for different vegetation

types that when temperature increases, spring onset is earlier and autumn senescence is later (Christensen et al., 2007; Linkosalo et al., 2009; Migliavacca et al., 2012; Morin et al., 2010; Post and Forchhammer, 2008), increasing the length of the growing season and the amount of CO2 that is uptake by ecosystems (Richardson et al., 2013). On the other hand, where the Tair regression coefficient is negative and the temperature increase during the first part of the growing season the speed of the water consumption will increase producing an earlier $DOY_{GPPmax}$ (Figure 8).





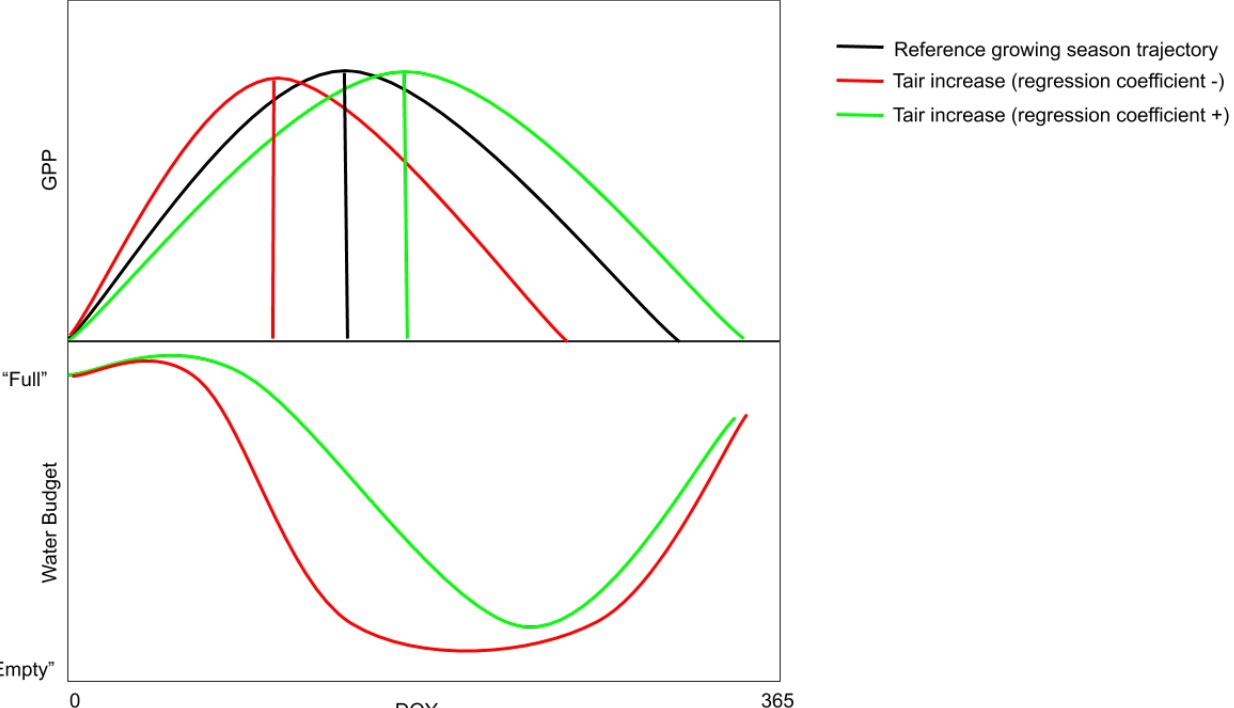

**Figure 8.** Theoretical relationship between the sign of air temperature (Tair) in the circular regression model and the water budget in an ecosystem. When the sign of the regression coefficient is negative and temperature increase the $DOY_{GPPmax}$ is reached earlier (Red), on the other hand if the sign is positive and temperature increase the $DOY_{GPPmax}$ is reach later (Green).





Ecosystems with two growing seasons per year represent a very interesting case of the effect of climate drivers on $DOY_{GPPmax}$ across different growing seasons. In Llano de los Juanes, Spain (ES-LJu, Figure 6) $DOY_{GPPmax}$ is reached in the first growing season when the rainy season is finishing, while in the second growing season $DOY_{GPPmax}$ is reached in the middle of the rainy season (Data not shown). The effect of temperature for the first growing season is positive suggesting that if we increase the temperature during the period before, the $DOY_{GPPmax}$ will happen later. Following our hypothesis mentioned above this will

happen because the speed of the water consumption is reduced, probably because the precipitation is also increase during the spring. However, as VPD has a negative effect and given the covariance between the Tair and VPD the effect of the increase of temperature is in part countered by the increase of VPD. During the second growing season the effect of Tair is negative meaning in this case that the water budget is lower, then if Tair increase the $DOY_{GPPmax}$ will be reached earlier.

       Phenology in Mediterranean ecosystems is mainly controlled by water availability (Kramer et al., 2000; Luo et al., 2018;

Peñuelas et al., 2009). However, our results suggest that $DOY_{GPPmax}$ is mainly sensitive to temperature. This result agrees with the analysis performed by (Gordo and Sanz, 2005) were the authors evaluated the phenological sensitivity of Mediterranean ecosystem to temperature and precipitation, and they concluded that temperature was the most important driver. Although water is a limiting factor in Mediterranean ecosystems, its influence on plant physiology and plant phenology can be completely different. In terms of physiology the GPPmax value can decrease but in terms of phenology $DOY_{GPPmax}$ can be still being the

same.

       Complex interactions between climate variables and phenological response and the interspecificity of the sensitivity at site level explain in part the poor power prediction of the model for grasslands, Evergreen Broadleaf Forest, and Deciduous Broadleaf Forests in the cross validation analysis (Figure 7). However, the power prediction for Mixed Forest and Evergreen Needleleaf Forests is good, also when the distribution of the latitudinal gradient is not the same for all the sites. These results

reflect that circular regression model can be extrapolated from different sites, to predict the $DOY_{GPPmax}$ interannual variability. This advantage could be a way to solve the common critic that phenological models can not be extrapolated generating only ad hoc hypothesis (Richardson et al., 2013).

## 5   Conclusions

In this study we explore the potential of "circular regressions" to explain the phenology of maximal CO2 uptake rates. We

conclude that 1) shortwave incoming radiation, and temperature are the main drivers of the timing of maximal CO2 uptake at global scale; precipitation and VPD only play a secondary role. 2) Although the sensitivity of the $DOY_{GPPmax}$ to the climate drivers is site specific, it is possible to extrapolate the circular regression model for different sites with the same vegetation type and similar latitudes. Finally, we demonstrated using simulated and empirical data, that circular regression produces more accurate results than linear regression, in particular in cases when data needs to be explored across hemispheres.



*Acknowledgements.* This project has received funding from the European Union's Horizon 2020 research and innovation programme via the Trustee project under the Marie Skłodowska-Curie grant agreement No 721995. This work used eddy covariance data acquired and shared by the FLUXNET community, including these networks: AmeriFlux, AfriFlux, AsiaFlux, CarboAfrica, CarboEuropeIP, CarboItaly, CarboMont, ChinaFlux, Fluxnet-Canada, GreenGrass, ICOS, KoFlux, LBA, NECC, OzFlux-TERN, TCOS-Siberia, and USCCC. The ERA-Interim re-analysis data are provided by ECMWF and processed by LSCE. The FLUXNET eddy covariance data processing and harmonization was carried out by the European Fluxes Database Cluster, AmeriFlux Management Project, and Fluxdata project of FLUXNET, with the support of CDIAC and ICOS Ecosystem Thematic Center, and the OzFlux, ChinaFlux and AsiaFlux offices.





## Appendix A: FLUXNET Sites

Table A1: FLUXNET sites used in our study. We report the name of the sites, time period used for the analysis, the climate class of each site following the Köppen-Geiger classification: Tropical monsoon climate (Am), Tropical savanna climate (Aw), Cold semi-arid climates (BSk), Humid subtropical climate (Cfa), Oceanic climate (Cfb), Hot-summer mediterranean climate (Csa), Warm-summer mediterranean climate (Csb), Humid subtropical climate (Cwa), humid continental climate (Dfb), Subarctic climate (Dfc, Dsc), and Tundra climate (ET). We also report the Vegetation type of the sites: Closed Shrublands (CSH), Deciduous Broadleaf Forests (DBF), Evergreen Broadleaf Forest (EBF), Evergreen Needleleaf Forests (ENF), Grasslands (GRA), Mixed Forests (MF), Open Shrublands (OSH), Savannas (SAV), Permanent Wetlands (WET), Woody Savannas (WSA).

| Site name | Köppen-Geiger class | Vegetation type | Period | N. years analyzed | Citation | Data DOI |
|---|---|---|---|---|---|---|
| US-Ha1 | Dfb | DBF | 1992:2012 | 21 | (Urbanski et al., 2007) | 10.18140/FLX/1440071 |
| US-PFa | Dfb | MF | 1996:2014 | 19 | (Berger et al., 2001) | 10.18140/FLX/1440089 |
| BE-Bra | Cfb | MF | 1999:2002, 2004:2014 | 15 | (Carrara et al., 2004) | 10.18140/FLX/1440128 |
| BE-Vie | Cfb | MF | 1997:2014 | 18 | (Aubinet et al., 2001) | 10.18140/FLX/1440130 |
| DE-Tha | Cfb | ENF | 1996:2014 | 19 | (GrüNwald and Bernhofer, 2007) | 10.18140/FLX/1440152 |
| DK-Sor | Cfb | DBF | 1996:2014 | 19 | (Pilegaard et al., 2011) | 10.18140/FLX/1440155 |
| FI-Hyy | Dfc | ENF | 1996:2014 | 19 | (Suni et al., 2003) | 10.18140/FLX/1440158 |
| IT-Col | Csa | DBF | 1996:2014 | 19 | (Valentini et al., 1996) | 10.18140/FLX/1440167 |
| NL-Loo | Cfb | ENF | 1996:2014 | 18 | (Moors, 2012) | 10.18140/FLX/1440178 |
| CH-Dav | ET | ENF | 1997:2014 | 18 | (Zielis et al., 2014) | 10.18140/FLX/1440178 |
| RU-Fyo | Dfb | ENF | 1998:2014 | 17 | (Kurbatova et al., 2008) | 10.18140/FLX/1440183 |
| US-NR1 | Dfc | ENF | 1999:2014 | 16 | (Monson et al., 2002) | 10.18140/FLX/1440087 |
| IT-Ren | Dfc | ENF | 1999, 2002:2003, 2005:2013 | 12 | (Montagnani et al., 2009) | 10.18140/FLX/1440173 |
| US-MMS | Cfa | DBF | 1999:2014 | 16 | (Schmid et al., 2000) | 10.18140/FLX/1440083 |





| US-WCr | Dfb | DBF | 1999:2006, 2011:2014 | 12 | (Curtis et al., 2002) | 10.18140/FLX/1440095 |
|---|---|---|---|---|---|---|
| CA-Man | Dfc | ENF | 1994:1996, 1998:2003 | 12 | (Brooks et al., 1997) | 10.18140/FLX/1440035 |
| DK-ZaH | ET | GRA | 2000:2010, 2012:2014 | 14 | (Lund et al., 2012) | 10.18140/FLX/1440224 |
| FR-Pue | Csa | EBF | 2000:2015 | 15 | (Rambal et al., 2004) | 10.18140/FLX/1440164 |
| US-Los | Dfb | WET | 2001:2008, 2010, 2014 | 10 | (Davis et al., 2003) | 10.18140/FLX/1440076 |
| US-UMB | Dfb | DBF | 2000:2014 | 15 | (Curtis et al., 2002) | 10.18140/FLX/1440093 |
| US-Var | Csa | GRA | 2001:2014 | 14 | (Xu and Baldocchi, 2004) | 10.18140/FLX/1440094 |
| AU-How | Aw | WSA | 2002:2014 | 13 | (Beringer et al., 2007) | 10.18140/FLX/1440125 |
| AU-Tum | Cfb | EBF | 2001:2014 | 14 | (Leuning et al., 2005) | 10.18140/FLX/1440126 |
| FI-Sod | Dfc | ENF | 2001:2014 | 14 | (Thum et al., 2007) | 10.18140/FLX/1440160 |
| IT-SRo | Csa | ENF | 1999:2012 | 14 | (Chiesi et al., 2005) | 10.18140/FLX/1440176 |
| US-Syv | Dfb | MF | 2001:2007, 2012:2014 | 10 | (Desai et al., 2005) | 10.18140/FLX/1440091 |
| US-Ton | Csa | WSA | 2001:2014 | 14 | (Xu and Baldocchi, 2003) | 10.18140/FLX/1440092 |
| ZA-Kru | Cwa | SAV | 2000:2005, 2007:2013 | 13 | (Archibald et al., 2009) | 10.18140/FLX/1440188 |
| DE-Hai | Cfb | DBF | 2000:2012 | 13 | (Knohl et al., 2003) | 10.18140/FLX/1440148 |
| FR-LBr | Cfb | ENF | 1996:2008 | 13 | (Berbigier et al., 2001) | 10.18140/FLX/1440163 |
| IT-Cpz | Csa | EBF | 2000:2008 | 9 | (Garbulsky et al., 2008) | 10.18140/FLX/1440168 |
| US-Me2 | Csb | ENF | 2002:2014 | 13 | (Treuhaft et al., 2004) | 10.18140/FLX/1440079 |
| IT-Lav | Cfb | ENF | 2003:2014 | 12 | (Marcolla et al., 2003) | 10.18140/FLX/1440169 |





| RU-Cok | Dsc | OSH | 2003:2013 | 11 | (Molen et al., 2007) | 10.18140/FLX/1440182 |
|--------|-----|-----|-----------|----|----------------------|----------------------|
| AT-Neu | Dfc | GRA | 2002:2012 | 11 | (Wohlfahrt et al., 2008) | 10.18140/FLX/1440121 |
| CH-Lae | Cfb | MF | 2004:2014 | 11 | (Etzold et al., 2011) | 10.18140/FLX/1440134 |
| DE-Gri | Cfb | GRA | 2004:2014 | 11 | (Prescher et al., 2010) | 10.18140/FLX/1440147 |
| GF-Guy | Am | EBF | 2004:2014 | 11 | (Bonal et al., 2008) | 10.18140/FLX/1440165 |
| IT-MBo | Dfb | GRA | 2003:2013 | 11 | (Marcolla et al., 2011) | 10.18140/FLX/1440170 |
| IT-Noe | Csa | CSH | 2004:2014 | 11 | (Marras et al., 2011) | 10.18140/FLX/1440171 |
| IT-Ro2 | Csa | DBF | 2002:2008, 2010:2012 | 10 | (Tedeschi et al., 2006) | 10.18140/FLX/1440175 |
| US-Blo | Csa | ENF | 1997:2007 | 11 | (Baker et al., 1999) | 10.18140/FLX/1440068 |
| US-GLE | Dfc | ENF | 2005:2014 | 10 | (McDowell et al., 2000) | 10.18140/FLX/1440069 |
| US-SRM | BSk | WSA | 2004:2014 | 11 | (Scott et al., 2008) | 10.18140/FLX/1440090 |
| US-Wkg | BSk | GRA | 2004:2014 | 11 | (Emmerich, 2003) | 10.18140/FLX/1440096 |
| BR-Sa1 | Am | EBF | 2002:2005, 2009:2011 | 7 | (Saleska et al., 2003) | 10.18140/FLX/1440032 |
| CH-Cha | Cfb | GRA | 2005:2014 | 10 | (Merbold et al., 2014) | 10.18140/FLX/1440131 |
| CH-Fru | Cfb | GRA | 2005:2014 | 10 | (Imer et al., 2013) | 10.18140/FLX/1440133 |
| ES-LJu | Csa | OSH | 2005:2013 | 9 | (Serrano-Ortiz et al., 2009) | 10.18140/FLX/1440226 |
| FR-Fon | Cfb | DBF | 2005:2014 | 10 | (Delpierre et al., 2016) | 10.18140/FLX/1440161 |
| CZ-wet | Cfb | WET | 2006:2014 | 9 | (Dušek et al., 2012) | 10.18140/FLX/1440145 |
| IT-Ro1 | Csa | DBF | 2001:2008 | 8 | (Rey et al., 2002) | 10.18140/FLX/1440174 |





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

*Author contributions.* DEPM, TM, MM, and MDM designed the study in collaboration with MR and CR. DEPM conducted the analysis and wrote the manuscript with substantial contributions from all co-authors

*Competing interests.* The authors declare that they have no conflict of interest