# Peer review of "Ecosystem physio-phenology revealed using circular statistics"

_Biogeosciences, 2019_

## Referee Comment (RC1) · Anonymous Referee #1 · 12 Nov 2019

In "Ecosystem physio-phenology revealed using circular statistics", Pabon-Moreno et al. have analyzed how the timing of maximum gross primary productivity is related to climate variables such as air temperature, solar radiation, precipitation, and VPD. They have analyzed 52 FLUXNET sites with more than 7 years of data and applied a circular regression method to (a) understand which environmental variable best predicts the timing of GPPmax and (2) measure the sensitivity of the response to each variable and (c) evaluate the method for different plant functional types. The topics is interesting, and the questions are relevant.

The authors have also performed a simulation analysis to compare linear and circular regression methods, in particular given that some of the sites are in the southern hemisphere and hence may not be on the same calendar year as the northern hemisphere

sites, the authors have justified circular regression methods are more appropriate than linear regression methods.

The manuscript is generally well written and presented, however I have a couple major concerns related to the methods and conclusion that I strongly recommend being addressed by the authors.

1) I am not sure how finding shortwave radiation is related to the annual trend of GPP is surprising. Especially with the not particularly high correlation values from the model outputs. My concern is that what the model predicts may be actually the average seasonality of the site, which is generally represented/regulated by the annual variation of solar radiation. I think it would have been more convincing if the model could predict "weird years" rather than normal years. So, one might argue that the model is tuned to track the seasonality of the sites with an average predictability power. See my next comment which is related.

2) My other concern is that DOY values were directly used in the model as response variables. However, to analyze the inter-annual variability of the response, the anomalies should be used in the model. This is somehow related to the previous comment, as using site-specific model and absolute response values may result in obtaining the average annual trend and not the year-to-year variabilities. I think it would be best if the authors could use anomalies for each site as "y" in equation 2. Note that using absolute values in a consolidated model (all sites together) is another potentially good idea but that would detect the spatial (or site-by-site) patterns in the data rather than the temporal trends (which is the main question here).

There are also a few minor comments that I came across:

1- There is extensive use of parenthesis in the paper that sometimes make the narrative hard to follow. I suggest avoiding unnecessary parentheses in the manuscript.

2- The authors have used present tense throughout the manuscript at many places

where past tense verbs are recommended.

3- line 141, "closed parenthesis" that was never opened

4- the narrative of the Results section can be improved, especially because the reader has to go back to the method to remember the terminologies and acronyms related to the method.

5- line 277: "Although the sensitivity of the DOYGPPmax to the climate drivers is site specific, it is possible to extrapolate the circular regression model for different sites with the same vegetation type and similar latitudes." That's a big claim. I'm not sure if the manuscript has provided convincing evidence with only 52 sites to support this.

6- What are the temporal windows for each predictor variable?

—-

I hope the authors find this review helpful in improving their manuscript.

---

## Referee Comment (RC2) · Anonymous Referee #2 · 26 Nov 2019

General comments: In the manuscript "Ecosystem physio-phenology revealed using circular statistics", Pabon-Moreno et al. used a new method – the circular linear regression to estimate the timing of the maximum gross primary productivity (DOYgppmax) at 52 eddy covariance towers, and further quantified the sensitivity of DOYgppmax to a range of climate variables based on the results from this new regression. The manuscript is relevant to the topics of the journal. While I agree with the authors that circular linear regression has the potential to be a framework of future generalized phenology models, I have some doubts about the advantages of circular regression over the conventional linear regression approach, as well as the interpretation of results. It may need some substantial revisions. I apologize that I cannot be more supportive at this stage. I hope the authors can find this review helpful (please see below).

[Figure]

Specific comments: Major specific comments: 1. The authors introduced two advantages of circular regression 1) it is more accurate than linear regression 2) it can analyze the phenological event regardless of the locations of events, esp. for the southern hemisphere. For 1), I am concerned about circular reasoning, as the authors used two phenological events pre-defined by circular regression to compare the performance of circular regression and linear regression, it is very likely the circular regression can outperform linear regression in this case. In addition, the author used the distance between observed beta and estimated beta to assess the efficiency of two models, and suggested that because the magnitude of distance for beta1 is larger than the distance for beta2, and the results on distance for beta1 favored circular regression, so circular regression is better. But the magnitude of distance for beta is also dependent on beta itself. Beta2 (0.3) is larger than beta1 (0), after normalize the distance of beta by beta, the result based on beta1 does not carry more weight than beta2, and the results on the distance of beta2 in fact favored linear regression. For 2), I am not sure why conventional phenology models cannot be used in the Southern Hemisphere (e.g. L208-209), say the degree-day model can be easily deployed if we know the temperature preceding budburst in Australia (e.g. Webb et al., 2008) and we can also get meaningful climate sensitivity of the event. Overall, I am not sure the circular model is superior to conventional models based on the evidence available in the manuscript.

2. Some questions over the interpretation of the results. First, I am a bit worried about overfitting of the model, as the leave-one-out validation suggest much less robust performance (r = -0.3 ∼ 0.7) for PFTs compared to the r (r = 0.7 ∼ 0.9 according to Table S1) we obtained using the training dataset. Second, at seasonal time scale, air temperature, radiation and VPD are all highly correlated with each other, how much can we trust their respective sensitivities estimated by circular regression. Wouldn't the sensitivity of air temperature be account for by the sensitivity of radiation if there is co-linearity between the two? Third, I guess the so-called "memory effect" or "accumulated effect" of past climate is considered in circular regression through equation (1). Is this potentially one of the key differences between circular model and linear

model? Does it mean the climate conditions closer to the event is more important than the climate conditions further back, and different climate variables are prescribed with different half-life here? I hope this part of the method is clearer. Fourth, the authors delegated the complex temperature sensitivity to consumption of available water (L240-). I am not sure there is a clear mechanistic underlying this link as there is no evidence supporting plant water uptake is related to temperature here. Soil water content may directly impact GPP (Stocker et al., 2018), it is not necessarily related to temperature, maybe VPD though. My major concern is about the robustness of the climate sensitivity identified in the manuscript.

Minor specific comments: 1. it is not accurate to say "(DOYgppmax)... is the time the plants reach their maximum potential for CO2 absorption". GPP is the product of vegetation density (i.e. LAI) and the photosynthesis of individual leaves. When leaves have the maximum photosynthetic capacity/potential, it does not mean the whole canopy would be the most productive, as leaf photosynthesis can be downregulated by environment, and it also depends on how many leaves are there in the ecosystem.

2. Figure 1. In figure caption and in the text (L64), you mentioned each line represents the interannual variability. I feel it needs further clarification on how to read the figure. From what I understand, the distance between the line and the circle indicate the frequency of DOYgppmax, and the spread of linear may imply the variability of DOYgppmax.

3. Method. Need more explanation about equation (1), as it not clear the meaning of x, N, N0, and the reason to include this half-life process here.

4. I think the title of the paper might overshoot what in fact was done in the paper, since only one type of phenological event was studied, and I am not sure there is a pattern that really is "revealed" here that we can easily extrapolate for us to understand DOYgppmax due to the reported site-specific sensitivities. The concept of physiphenology is new to me, maybe the authors can provide a reference? I feel most conventional phenological events (e.g. budburst, leafout, leaf coloring, leaf senescence) are physiological changes of plants, so why they are not qualified as physi-phenology or do we really need this definition here. DOYgppmax sounds like a carbon uptake phenological phase.

Technical comments: 1. "2" in "CO2" is subscript 2. please define "GPPmax" at its first appearance. 3. L201, according to Figure 7, GRA is -0.3 rather than 0? 4. How to interpret the tendency in Figure 7? 5. L150, "leaf" to "leave" 6. Table A1, maybe list the site according to their names or vegetation types. Now it is based on doi and not easy for readers to search sites. 7. It would be helpful to condense figures in supplementary material 2 into a table, showing the sensitivity of each climate variable and significant level indicated by *. And please consider merging two supplementary materials into one.

---

## Author Comment (AC1) · 15 Feb 2020

**Ecosystem physio-phenology revealed using circular statistics**

**Response to reviewers**

Dear Editors,

Please find below our responses to the reviewers. The comments by the reviewers are very relevant and will certainly help us to improve the quality of the manuscript. In the following we repeat the comments by the reviewers in bold and our response (RS) to each one in normal font.

**Responses to Reviewer 1**

**Reviewer 1: In "Ecosystem physio-phenology revealed using circular statistics", Pabon-Moreno et al. have analyzed how the timing of maximum gross primary productivity is related to climate variables such as air temperature, solar radiation, precipitation, and VPD. They have analyzed 52 FLUXNET sites with more than 7 years of data and applied a circular regression method to (a) understand which environmental variable best predicts the timing of GPPmax and (2) measure the sensitivity of the response to each variable and (c) evaluate the method for different plant functional types. The topics is interesting, and the questions are relevant. The authors have also performed a simulation analysis to compare linear and circular regression methods, in particular given that some of the sites are in the southern hemisphere and hence may not be on the same calendar year as the northern hemisphere sites, the authors have justified circular regression methods are more appropriate than linear regression methods. The manuscript is generally well written and presented, however I have a couple major concerns related to the methods and conclusion that I strongly recommend being addressed by the authors.**

We thank the reviewer for the accurate summary of our paper.

**I am not sure how finding shortwave radiation is related to the annual trend of GPP is surprising. Especially with the not particularly high correlation values from the model outputs. My concern is that what the model predicts may be actually the average seasonality of the site, which is generally represented/regulated by the annual variation of solar radiation. I think it would have been more convincing if the model could predict "weird years" rather than normal years. So, one might argue that the model is tuned to track the seasonality of the sites with an average predictability power. See my next comment which is related.**

The reviewer is right that shortwave incoming radiation (SWin) driving GPP - confirming this would be indeed no surprise. But please note that this paper does not analyze GPP: we are predicting the timing when GPP is maximized (the units we predict are "day of year" and not "g C /( m2 day)" ). GPPmax represents the "optimum" ecosystem state when ecosystems are maximizing the uptake of $CO_2$ per year. We would also clarify that maximizing "predictability" is not our main aim as we are primarily interested in understanding the sensitivity of this state of ecosystem physio-phenology to climate variability. Given that radiation typically has a very low interannual variability we expect that the timing of GPPmax should be sensitive to other factors.

**My other concern is that DOY values were directly used in the model as response variables. However, to analyze the inter-annual variability of the response, the anomalies should be used in the model. This is somehow related to the previous comment, as using site-specific model and absolute response values may result in obtaining the average annual trend and not the year-to-year variabilities. I think it would be best if the authors could use anomalies for each site as "y" in equation 2.**

Equation 2 describes the circular linear regression where μ is the mean angular direction of a Von Mises distribution. As we mentioned in line 108 the mean angular direction is estimated via maximum likelihood. All interannual observations of $DOY_{GPPmax}$ are used on the model, and the final result is constrained to a Von Mises distribution. The μ parameter cannot be removed from the equation, on the other hand the anomalies are considered into the amplitude of the Von Mises distribution (β) that is estimated internally.

**Note that using absolute values in a consolidated model (all sites together) is another potentially good idea but that would detect the spatial (or site-by-site) patterns in the data rather than the temporal trends (which is the main question here).**

The use of DOY values is necessary to quantify the sensitivity to the climate variables. On the other hand, if only consider $DOY_{GPPmax}$ anomalies (outliers) the main research question regarding climate sensitivity can not be solved given that we will not analyze a representative sample of the observations. Considering only $DOY_{GPPmax}$ outliers the research question should be more related to extreme events or temporal anomalies that as we mentioned in the previous comment are not the main topic in our study.

**Minor comments:**

**There are also a few minor comments that I came across:**

**1- There is extensive use of parentheses in the paper that sometimes make the narrative hard to follow. I suggest avoiding unnecessary parentheses in the manuscript.**

The paper will be revised accordingly.

**2- The authors have used present tense throughout the manuscript at many places where past tense verbs are recommended.**

This is a matter of "style" and we would like to keep the writing in present as we feel it is better to read.

**3- line 141, "closed parenthesis" that was never opened**

Solved. The parenthesis was removed

**4- the narrative of the Results section can be improved, especially because the reader has to go back to the method to remember the terminologies and acronyms related to the method.**

Thank you for this observation, we will revise the manuscript accordingly.

**5- line 277: "Although the sensitivity of the DOYGPPmax to the climate drivers is site specific, it is possible to extrapolate the circular regression model for different sites with the same vegetation type and similar latitudes." That's a big claim. I'm not sure if the manuscript has provided convincing evidence with only 52 sites to support this.**

Given that we used cross-validation to measure the performance of the model per vegetation type. We consider 52 sites should be enough to provide a robust analysis. On the other hand, 52 sites are the data available globally with at least 7 years of records.

**6- What are the temporal windows for each predictor variable?**

In our study the temporal window for the predictors is given by the half-time parameter of the half-life decay function (See Supplement 1. Half-time sensitivity analysis (System memory to explain $DOY_{GPPmax}$)). In this section we run a sensitivity analysis to quantify how the change of the half-time parameter affects the correlation coefficient between the observed and predicted $DOY_{GPPmax}$.

**Reviewer 2**
**General comments: In the manuscript "Ecosystem physio-phenology revealed using circular statistics", Pabon-Moreno et al. used a new method – the circular linear regression to estimate the timing of the maximum gross primary productivity (DOYgppmax) at 52 eddy covariance towers, and further quantified the sensitivity of DOYgppmax to a range of climate variables based on the results from this new regression. The manuscript is relevant to the topics of the journal. While I agree with the authors that circular linear regression has the potential to be a framework of future generalized phenology models, I have some doubts about the advantages of circular regression over the conventional linear regression approach, as well as the interpretation of results. It may need some substantial revisions. I apologize that I cannot be more supportive at this stage. I hope the authors can find this review helpful (please see below)**

We thank the reviewer for sharing her considerations. The comments provided below are indeed very helpful.

**The authors introduced two advantages of circular regression 1) it is more accurate than linear regression 2) it can analyze the phenological event regardless of the locations of events, esp. for the southern hemisphere. For 1), I am concerned about circular reasoning, as the authors used two phenological events pre-defined by circular regression to compare**

the performance of circular regression and linear regression, it is very likely the circular regression can outperform linear regression in this case. In addition, the author used the distance between observed beta and estimated beta to assess the efficiency of two models, and suggested that because the magnitude of distance for beta1 is larger than the distance for beta2, and the results on distance for beta1 favored circular regression, so circular regression is better. But the magnitude of distance for beta is also dependent on beta itself. Beta2 (0.3) is larger than beta1 (0.1), after normalize the distance of beta by beta, the result based on beta1 does not carry more weight than beta2, and the results on the distance of beta2 in fact favored linear regression.

Regarding the first point. We used equation 2 to simulated the data. Nevertheless we are analyzing the performance to recover the original beta values of the equation and not the predictive power of the model. We used equation 2 given that linear regression does not allow to define the mean timing of phenological events. This is problematic especially when we want to analyze phenological events at the beginning and the end of the year.

Regarding the second point there is a misunderstanding related to the beta values. In our study $beta_1 = 0.3$ and $beta_2 = 0.1$ (Line 124). For this reason if we divide the distance by the beta values as suggested by the reviewer, the tendency of the results does not change (Please see the plots below). In both plots, circular regression has a better performance recovering $beta_1$, while linear regression has a better performance recovering $beta_2$ when the number of data is greater than 100. We will include the plot with the distances divided by the beta value to show that there is not a strong effect in the results. We will modify the line 135 of the methods by: "We estimate the difference between the recovered and the original coefficient divided by the beta value as the efficiency of the model (i.e. lower values mean higher efficiency).". And we will modify the line 166 of the results by: "Nevertheless, the differences between both regressions for $beta_2$ are of the order of 0.2 while the differences for $beta_1$ are of the order of 0.5."

Original Plot:

[Figure]

Plot normalizing the values per beta (distance / beta):

[Figure]

**For 2), I am not sure why conventional phenology models cannot be used in the Southern Hemisphere (e.g. L208-209), say the degree-day model can be easily deployed if we know the temperature preceding budburst in Australia (e.g. Webb et al., 2008) and we can also get meaningful climate sensitivity of the event. Overall, I am not sure the circular model is superior to conventional models based on the evidence available in the manuscript.**

Please note that the only claim we make is that circular regression is more suitable than conventional linear models for analyzing phenological data - of course, process-based phenological models should outperform such statistical approaches. But our analysis reveals that we can learn the sensitivity to climate drivers in a purely empirical manner. In general, in any degree-day model there is a parameter to set an initial time to start accumulating warming. This will require again to define a t0 and in our view circular statistics could potentially avoid manual tunings of this kind.

**Some questions over the interpretation of the results. First, I am a bit worried about overfitting of the model, as the leave-one-out validation suggest much less robust performance (r = -0.3 ~ 0.7) for PFTs compared to the r (r = 0.7 ~ 0.9 according to Table S1) we obtained using the training dataset.**

As we mentioned in previous comments the main objective of the study was to analyze the sensitivity of $DOY_{GPPmax}$ to different climate drivers. For this reason, each site has a unique "r" we consider that high r values are not an argument for dismissing the sensitivities of the climate variables. After cross-validation is expected that the predictive power of the model decreases, but the performance is not so bad considering it is estimated across vegetation types.

**Second, at seasonal time scale, air temperature, radiation and VPD are all highly correlated with each other, how much can we trust their respective sensitivities estimated by circular regression. Wouldn't the sensitivity of air temperature be account for by the sensitivity of radiation if there is co-linearity between the two?**

We performed a variance inflation factor analysis (VIF) for all sites-variables. The analysis shows (see plot below) that the colinearity of Air temperature, Shortwave Incoming Radiation, and VPD increases the variance of the regression coefficient. (VIF > 5). To solve this problem it is necessary to implement a PCA with these variables and run again all the analysis using the first axis of the PCA and precipitation as predictors of $DOY_{GPPmax}$. This means a major change in

the manuscript given that the results of the sensitivity of $DOY_{GPPmax}$ to climate variables will change. The respective discussion, and the conclusion need to be re-written. The revised version of the manuscript will contain these changes.

[Figure]

**Third, I guess the so-called "memory effect" or "accumulated effect" of past climate is considered in circular regression through equation (1). Is this potentially one of the key differences between circular model and linear model? Does it mean the climate conditions closer to the event is more important than the climate conditions further back, and different climate variables are prescribed with different half-life here? I hope this part of the method is clearer.**

We will add a better explanation:
"The idea of the decay function is that events in the present ($DOY_{GPPmax}$) are affected by past conditions (past climatic conditions). In this sense, the climatic conditions when $DOY_{GPPmax}$ happens will have a weight of 1 to explain it. The day before will be less than the first day (e.g. weight of 0.8) and so on."

**Fourth, the authors delegated the complex temperature sensitivity to consumption of available water (L240-). I am not sure there is a clear mechanistic underlying this link as there is no evidence supporting plant water uptake is related to temperature here. Soil water content may directly impact GPP (Stocker et al., 2018), it is not necessarily related to temperature, maybe VPD though. My major concern is about the robustness of the climate sensitivity identified in the manuscript.**

The relationship temperature ~ water consumption is a hypothesis that we put forward to explain the non-predominant sign for the temperature coefficient. It is important to mention that GPPmax is different to $DOY_{GPPmax}$. The last one is the timing when GPP is maximized during the growing season. In this sense, the magnitude of GPP can decrease when soil water content decreases but this will not necessarily affect $DOY_{GPPmax}$. To clarify this point we will modify the legend of figure 8 from "theoretical" to "hypothetical"

**Minor specific comments:**

**1. it is not accurate to say "(DOYgppmax). . . is the time the plants reach their maximum potential for CO2 absorption". GPP is the product of vegetation density (i.e. LAI) and the photosynthesis of individual leaves. When leaves have the maximum photosynthetic capacity/potential, it does not mean the whole canopy would be the most productive, as leaf photosynthesis can be downregulated by environment, and it also depends on how many leaves are there in the ecosystem.**

The reviewer is right that our wording is not very accurate here. We will clarify that we are analyzing a metric at the canopy scale. We will write "the time when the ecosystem reaches its maximum potential for CO2 absorption".

**2. Figure 1. In figure caption and in the text (L64), you mentioned each line represents the interannual variability. I feel it needs further clarification on how to read the figure. From what I understand, the distance between the line and the circle indicate the frequency of DOYgppmax, and the spread of linear may imply the variability of DOYgppmax.**

We will modify the legend to "The distance between the color line and the circle represents the frequency of the $DOY_{GPPmax}$ observations. The distance between the end and the beginning of the distribution represent the $DOY_{GPPmax}$ interannual variability"

**3. Method. Need more explanation about equation (1), as it not clear the meaning of x, N, N0, and the reason to include this half-life process here.**

We will modify the manuscript accordingly.

**4. I think the title of the paper might overshoot what in fact was done in the paper, since only one type of phenological event was studied, and I am not sure there is a pattern that really is "revealed" here that we can easily extrapolate for us to understand DOYgppmax due to the reported site-specific sensitivities. The concept of physiphenology is new to me, maybe the authors can provide a reference? I feel most conventional phenological events (e.g. budburst, leafout, leaf coloring, leaf senescence) are physiological changes of plants, so why they are not qualified as physi-phenology or do we really need this definition here. DOYgppmax sounds like a carbon uptake phenological phase.**

We defined ecosystem physio-phenology as the temporal variability of optimum and basal ecosystem states in terms of the exchange of energy and matter between the ecosystem and the atmosphere. We defined $DOY_{GPPmax}$ as a physio-phenological event because data is derived from the fluxes of the exchange of energy and matter between the ecosystems and the atmosphere and represents in a very accurate way the plants' photosynthesis. Budburst, leaf coloring etc. are phenological state of plants that in specific cases not necessarily represent the physiological state of the plants (e.g. A green canopy does not necessarily mean that plants are photosynthetically active during winter). This limit between how much we can extrapolate between physiology and the light reflectance of the leaves is surpassed by the eddy covariance technique allowing us to quantify the ecosystem fluxes.

Although we only analyze one physio-phenological event ($DOY_{GPPmax}$), given that this study introduced the conceptual and methodological framework to analyze physio-phenological events we consider that the title is according to the research presented in the paper.

Regarding the comment: "I am not sure there is a pattern that really is "revealed" here that we can easily extrapolate for us to understand DOYgppmax due to the reported site-specific

sensitivities". In section 4.2 "Sensitivity of $DOY_{GPPmax}$ to climate variables," we summarized the effect of each climate variable at global scale.

**Technical comments:**

**1. "2" in "CO2" is subscript**
Fixed

**2. please define "GPPmax" at its first appearance.**
Fixed. Line 46

**3. L201, according to Figure 7, GRA is -0.3 rather than 0?**
Fixed

**4. How to interpret the tendency in Figure 7?**
In Figure 7 the tendency (blue line) represents the overestimation or underestimation of the model for specific $DOY_{GPPmax}$ values.

**5. L150, "leaf" to "leave"**
Fixed

**6. Table A1, maybe list the site according to their names or vegetation types. Now it is based on doi and not easy for readers to search sites.**

We will modify the table showing the sites names by alphabetical order.

**7. It would be helpful to condense figures in supplementary material 2 into a table, showing the sensitivity of each climate variable and significant level indicated by \*. And please consider merging two supplementary materials into one.**

Given that $DOY_{GPPmax}$ sensitivity to the climate variables was estimated implementing bootstrapping, we consider that it is more important to show the distribution of the data than just the mean, also for the p-values. Regarding the second comment, we would like to keep the supplementary materials as separate.

---

## Author Response (AR2)

**Ecosystem physio-phenology revealed using circular statistics**

**Changes in the manuscript**

**12-06-2020**

Dear Editor,

Please find below our responses to the reviewers, a description of the major changes, and a detailed version of the manuscript with all the changes at the end of the document.

In the following, we repeat the **comments by the reviewers in bold** and our response (RS) to each one in normal font:

Responses to Reviewer 1:

**Reviewer 1: In the manuscript entitled "Ecosystem physio-phenology revealed using circular statistics" submitted to Biogeosciences by Pabon-Moreno et al, the authors have performed an analysis of the timing of GPPmax across 52 flux sites using a circular regression method. The authors have also compared circular regression and linear regressing using simulated data and concluded the circular regression may be more suitable for these kinds of analysis.**

**Although the work is not of its first kind on the topic of circular stats for phenological modeling -as the authors have also pointed this out-, but the manuscript is still interesting. I believe the presentation of the work, however, can be much improved. There are also important questions and ambiguity about some of the technical parts of the manuscript that are not currently clear. In the manuscript, there are assumptions without proper justifications or performing sensitivity analysis for those assumptions. The presentations of the figures could also be improved. Overall, there is a lot of room for improvement.**

We thank the reviewer for the comments. We agree that circular statistics have been used earlier for phenological analysis. However, in the specific context addressed in our paper (using ecosystem scale metrics inferred from eddy covariance towers) this is new, and we can provide some novel insights. We also would like to kindly point out that the comparison of circular vs. linear regressions is only one part of the entire work. The main research question is dedicated to understanding how climate conditions affect $DOY_{GPPmax}$ - and the tool to address questions of this kind (circular regression) needs to be introduced properly. Regarding the technical aspects we solved all the doubts in the following questions. We kindly point out that the sensitivity analysis for the decay function was performed and is shown in Supplement 1. We clarify this in the manuscript. We also performed a new sensitivity analysis for the comparison between the circular and linear regression using a range of regression coefficients as suggested by the reviewer. The detailed changes are presented at the end of the manuscript.

**- L18-19: the last sentence of the abstract is suggested to be rephrased, it kind of awkward.**

We apologize for the awkward wording in the submitted draft which was: "In particular global analyses can benefit from this approach, i.e. when phase shifts play a role or double peaked growing seasons have to be considered." The sentence has now been rewritten as follows: "The analysis of phenological events at global scale can benefit from the use of circular statistics. Such an approach yields substantially more robust results for analyzing phenological dynamics in regions characterized by two growing seasons per year, or when the phenological event under scrutiny occurs between two years (i.e. $DOY_{GPPmax}$ in the Southern Hemisphere)".

**- L22: remove "e.g."**

We remove the "e.g" and now the paragraph reads:

"Phenology is the study of the timing of biological events that can be observed either at the organismic level or at the ecosystem scale (Lieth, 1974). For the latter, phenology is the study of some integral behavior across phenological states of the integrated canopy reflectance captured by remote sensing (Richardson et al., 2009; Zhang et al., 2003), or vegetation-driven ecosystem-atmosphere CO2-exchange fluxes (Richardson et al., 2010)."

**- L 45-48: "Bauerle et al. (2012) studied how photoperiod and temperature influence plants photosynthetic capacity, reporting that the photoperiod explains the variability of photosynthetic capacity better than temperature." Should mention study area, this is not a general fact, as it is a debated topic.**

We thank the reviewer for the comment. We added the number of species and the area of this study.

Now the manuscript between line 45 and line 47 reads:

"Bauerle et al. (2012) studied how photoperiod and temperature influence plants photosynthetic capacity for 23 tree species in temperate deciduous hardwoods, reporting that the photoperiod explains the variability of photosynthetic capacity better than temperature."

**-L74-78: What about the issue regarding southern vs northern hemisphere? earlier in the Introduction you discussed about it.**

We thank the reviewer for pointing this out.It is correct that this aspect needs to be addressed here. However, we tone it a bit down given that the overarching research question is on understanding how climate variability affects $DOY_{GPPmax}$. We have rewritten the final part of the introduction as follows:

"In this paper, we aim to identify the factors controlling the timing of the maximal seasonal GPP ($DOY_{GPPmax}$). The questions we want to answer are: First, can circular statistics describe and predict $DOY_{GPPmax}$ per vegetation type? This aspect requires testing the methodological advantages and caveats of circular statistics across hemispheres in comparison with linear methods. Second, can DOYGPPmax be explained using cumulative climate conditions? This question needs to consider different possibilities for generating temporally integrating features. And third, how is $DOY_{GPPmax}$ affected by the climatic conditions during the growing season? The last question requires a global cross-site analysis. Based on the findings of these three questions we then discuss the potential of circular regressions beyond this specific application case in related phenological problems and outline future applications.

**- L93: define i and tau, right after the equation.**

We re-wrote the description of the equation including the definition of i and tau. Now it reads:

" We aggregate the original times-series of the Tair, SWin, Precip, and VPD for each $DOY_{GPPmax}$ using a half-life decay function (eq. 1),

$$\langle \mathbf{x}_t \rangle = \frac{\sum_{i=0}^{\tau} x_{t-i} w_i}{\sum_{i=0}^{\tau-1} w_i} \quad (1)$$

for estimating an exponentially weighted mean of the observation vector, $\mathbf{x}_t = (x_t, x_{t-1}, \ldots, x_{t-\tau})^T$, at time step $t$. The symbol $\langle \cdots \rangle$ denotes the weighted average; $i$ indicates the number of days before $t$ going back up to $\tau = 365$ days.

**- L96: Is w0=1? If so, you need to tell the readers.**
Yes, we added w0 = 1 at line 100.

**- L98: Not clear why you chose a decaying weight relationship, there should be a sensitivity analysis on how the results are affected by different weights. Your assumption is simply not justified.**

We kindly point the reviewer to the supplement 1 where we documented a sensitivity analysis to identify the optimum decay weight per site as proposed here. To clarify this part, we re-wrote the paragraph that now reads:

"We perform a sensitivity analysis evaluating the effect of the half-time parameter and identify the optimum as the value when the variance explained by the circular regression model is maximum. The results are presented in Supplement 1"

**- L109: Based on equation (2), y is the phase (angle) not DOY as it is introduced in the next line. You need a coefficient to transform.**

To clarify this point, we added to the paragraph: "where $y$ is the target variable (i.e. $DOY_{GPPmax}$) in radians", and we added at the beginning of the section:

"Since circular response variable must be in radians or degrees. We transform the days of the year to radians using equation 3. For leap years we remove the last day."

$$rad = DOY \ \frac{360}{365} \ \frac{\pi}{180}$$

**- L127: replace "artificial" with "simulated"**

Yes, indeed.

**- L128: Again, your choices of β1 = 0.3 and β2 = 0.1 are arbitrary, you need to do a sensitivity analysis and test the results for ranges of β1 and β2.**

We thank the reviewer for this comment. To address this criticism we performed a completely new sensitivity analysis for a range of beta parameters values. We re-wrote the methods and result sections and introduced a new figure 3 in the submitted manuscript. At the end of the document we present the detailed changes of each section. In summary our results suggest that circular regressions can recover predefined coefficients in a set of simulations that represent realistic data with higher accuracy and precision than linear regressions. As was demonstrated with the previous analysis but this time, the evaluation of different beta regression coefficients will clarify that it's not a local statistical artifact but even more a general tendency. One more time, thanks to the reviewer for suggesting the analysis.

[Figure]

Figure 3. Accuracy and precision of linear and circular regression models by recovering the original regression coefficients of a circular regression. Left side: μ= 0(Maximum at the beginning of the year). Right side μ=pi(Maximum at Mid-year). a. and b. correspond to the differences in accuracy between the models. c. and d. correspond to the differences in the precision between the models. The blue color means better performance of the circular model compared with the linear model, and red color means higher performance of the linear model.

Figure 3 shows that for μ= 0 ($DOY_{GPPmax}$ at the beginning of the year) circular regression has a higher accuracy and precision than the linear regression for the entire space of regression coefficient values, with a maximum difference in the order of 0.1 for the accuracy, and the order of 1 for the precision. For μ=π ($DOY_{GPPmax}$ mid-year) linear model has a higher accuracy in most of the evaluated space with a maximum difference in the order of 0.001 compared with the circular regression. While, circular regression has a higher precision for most of the regression coefficients in the order of 0.001. These results show that circular regression has a higher precision to recover the original regression coefficients than linear regression no matter the moment of the year. On the other hand, circular regression has a higher accuracy than linear model at the beginning of the year. While at mid-year when linear is better the differences are in the order of 0.001.

**- Figure 2: not a great figure. very difficult to follow, what is 6.697? I would redraw with a better idea.**

We apologize for the lack of readability. Please note that 6.697 is the equal to -0.413 radians, we clarify this adding to the legend: "(The equivalent of -0.413 radians is 6.697. It is shown below the equation)". For us, the figure is important to show the effect of the increase in the variable when the regression coefficient is positive and negative. For this reason, we decided to include the equations with the plot and use the color to denote the increase.

**- Figure 4: in (a) the circles are smashed to together so the trends are not very clear.**

The transparency of the point can make the figure does not clear. With the use of density function for the distribution the outliers cannot be appreciated. Therefore, we would like to keep this figure. We also fixed the arrows in the figure because the counter parameter was setting wrongly before and the arrows were not showing the mean angular direction of the original data.

[Figure]

Figure 4. Correlation coefficient between the observed and predicted $DOY_{GPPmax}$ using climatic variables. Two sites are presented: a. US-Ha1, and b. AU-How. The observed $DOY_{GPPmax}$ (Green) is compared with the data retrieved using Circular (Orange) and Linear (Purple) regressions. Two correlation coefficients are used: Jammalamadaka-Sarna (JS) and Pearson product-moment (Pearson). In the circular plot the months and the day of the year (DOY) are also plotted every 75 days. The green arrow indicates the mean angular direction of the original data distribution.

**- L202: What is "the power of the prediction"? vague term.**

We rewrote this term as "predictive power". That is a well-established term to describe the property of a mathematical model to predict a specific event.

**- Figure 7: Another figure to be improved. The smoothed curves do not add anything. It is not even clear how they are obtained or if there is any quantitative metric to see the fit. See EBF for instance. I would get rid of them. Also, the plots must have square aspect ratio.**

We thank the reviewer for the comment. We removed the tendency lines to avoid confusion, replotted using the square aspect ratio and moved the JS correlation coefficient to the title of each plot.

[Figure]

Figure 7. Cross validation of the circular regression model to predict $DOY_{GPPmax}$ for different vegetation types using air temperature, short-wave incoming radiation, precipitation and vapor pressure deficit (see methods). Deciduous Broadleaf Forest (DBF). Evergreen BroadleafForest (EBF). Grassland (GRA). Mixed Forest (MF), and Evergreen Needleleaf Forest (ENF). For each vegetation type the Jammalamadaka-Sarna (JS) correlation coefficient is shown in the title of each plot. The red line represents the perfect fit.

**- L214: The sentence referred to a paper from 2013, and "concluded that phenology models need to be improved". In fact, there has been much improvements on phenology models since then.**

We thank the reviewer for the comment. We re-wrote this paragraph (between lines 247 and 253) and now it reads:

"Different phenological models have been developed ranging from empirical approaches (Richardson et al., 2013) to process models (Asse et al., 2020) over the last decades. As we demonstrate here, circular statistics opens new opportunities to increase the robustness of phenological models allowing to analyze ecosystems across hemispheres within the same consistent frame-work. In fact, the results on phenological sensitivity of $DOY_{GPPmax}$ indicate the complexity of ecosystem responses to climate variability. Our approach is a motivation towards integrating circular regressions into more complex statistical techniques like regression trees, Gaussian process, or artificial neural networks, targeting a circular response variable."

**- L217-218: "Indeed we considered our approach as a first step to implement more complex statistical techniques like decision trees, Gaussian process, or artificial neural networks, targeting a circular response variable." But the authors have not showed us how. From the manuscript, the connection between the circular regression and other methods mentioned here is not clear.**

We re-wrote this phrase and now it reads:

"Our approach is a motivation towards integrating circular regressions into more complex statistical techniques like regression trees, Gaussian process, or artificial neural networks, targeting a circular response variable"

We include this phrase as an outlook, to clarify that even if circular methods contribute to understanding phenological events, there is a lot of space where models can be improved. And as a motivation to the reader to keep contributing in this field.

**Major Changes**

We re-wrote the methods and results of the section 2.3 Circular vs. Linear Regression; the new version is presented below and in the new version of the manuscript.

[revised manuscript text omitted]